# Different dose regimens of a SARS-CoV-2 recombinant spike protein vaccine (NVX-CoV2373) in younger and older adults: A phase 2 randomized placebo-controlled trial

Neil Formica * , Raburn Mallory , Gary Albert , Michelle Robinson , Joyce S. Plested, Iksung Cho, Andreana Robertson , Filip Dubovsky, Gregory M. Glenn, for the 2019nCoV-101 Study Group[¶]

Novavax, Inc., Gaithersburg, Maryland, United States of America

¶ Membership of the 2019nCoV-101 Study Group is provided in the Acknowledgments.
* nformica@novavax.com

**Data Availability Statement:** All relevant data are within the manuscript and its Supporting Information files.

## Abstract

### Background

NVX-CoV2373 is a recombinant severe acute respiratory coronavirus 2 (rSARS-CoV-2) nanoparticle vaccine composed of trimeric full-length SARS-CoV-2 spike glycoproteins and Matrix-M1 adjuvant.

### Methods and findings

The phase 2 component of our randomized, placebo-controlled, phase 1 to 2 trial was designed to identify which dosing regimen of NVX-CoV2373 should move forward into late-phase studies and was based on immunogenicity and safety data through Day 35 (14 days after the second dose). The trial was conducted at 9 sites in Australia and 8 sites in the United States. Participants in 2 age groups (aged 18 to 59 and 60 to 84 years) were randomly assigned to receive either 1 or 2 intramuscular doses of 5-μg or 25-μg NVX-CoV2373 or placebo, 21 days apart. Primary endpoints were immunoglobulin G (IgG) anti-spike protein response, 7-day solicited reactogenicity, and unsolicited adverse events. A key secondary endpoint was wild-type virus neutralizing antibody response. After enrollment, 1,288 participants were randomly assigned to 1 of 4 vaccine groups or placebo, with 1,283 participants administered at least 1 study treatment. Of these, 45% were older participants 60 to 84 years. Reactogenicity was predominantly mild to moderate in severity and of short duration (median <3 days) after first and second vaccination with NVX-CoV2373, with higher frequencies and intensity after second vaccination and with the higher dose. Reactogenicity occurred less frequently and was of lower intensity in older participants. Both 2-dose regimens of 5-μg and 25-μg NVX-CoV2373 induced robust immune responses in younger and older participants. For the 2-dose regimen of 5 μg, geometric mean titers (GMTs) for IgG anti-spike protein were 65,019 (95% confidence interval (CI) 55,485 to 76,192) and 28,137 (95% CI 21,617 to 36,623) EU/mL and for wild-type virus neutralizing antibody (with an

**Funding:** Funding support for the study was provided from the Coalition of Epidemic Preparedness Innovations (CEPI) (https://cepi.net) to Novavax Inc. and not to any individual authors. The funders had no role in study design, data collection and analysis, decision to publish, or preparation of the manuscript.

**Competing interests:** I have read the journal's policy and the authors of this manuscript have the following competing interests: Authors RM, GA, JSP, IC, AR, FD, MR and GG all state to be employees and/or stock owners at Novavax. Dr. NF reports through my personal consulting company vehicle I consult for a variety of pre and post-licensure clinical development and medical affairs activities for other vaccine pharmaceutical and biotech companies. In the area of coronavirus vaccines I have a current consulting contract for post-licensure COVID vaccine support for approval and implementation of rollout of the AstraZeneca/Oxford COVID vaccine in Australia and New Zealand. Dr. GG reports grants from CEPI during the conduct of the study; personal fees and other from Novavax, grants from Bill and Melinda Gates Foundation, grants from Department of Defense, grants from Operation Warp Speed, other from RA Capital outside the submitted work.

**Abbreviations:** ADE, antibody-dependent enhancement; AE, adverse event; CI, confidence interval; COVID-19, coronavirus disease 2019; ELISA, enzyme-linked immunosorbent assay; FDA, Food and Drug Administration; GMFR, geometric mean fold rise; GMT, geometric mean titer; hACE2, human angiotensin-converting enzyme 2; IgG, immunoglobulin G; LLOQ, lower limit of quantification; MN, microneutralization; PBMC, peripheral blood mononuclear cell; PCR, polymerase chain reaction; PIMMC, potential immune-mediated medical condition; rSARS-CoV-2, recombinant severe acute respiratory coronavirus 2; SAE, serious adverse event; SARS-CoV-2, severe acute respiratory syndrome coronavirus 2.

inhibitory concentration of 50%—$MN_{50\%}$) were 2,201 (95% CI 1,343 to 3,608) and 981 (95% CI 560 to 1,717) titers for younger and older participants, respectively, with seroconversion rates of 100% in both age groups. Neutralizing antibody responses exceeded those seen in a panel of convalescent sera for both age groups. Study limitations include the relatively short duration of safety follow-up to date and current lack of immune persistence data beyond the primary vaccination regimen time point assessments, but these data will accumulate over time.

## Conclusions

The study confirmed the phase 1 findings that the 2-dose regimen of 5-μg NVX-CoV2373 is highly immunogenic and well tolerated in younger adults. In addition, in older adults, the 2-dose regimen of 5 μg was also well tolerated and showed sufficient immunogenicity to support its use in late-phase efficacy studies.

## Trial registration

ClinicalTrials.gov NCT04368988.

---

## Author summary

### Why was this study done?

- This study was designed to identify which dosing regimen of NVX-CoV2373 (1 or 2 doses of either 5 μg or 25 μg) was immunogenic and associated with a safety and tolerability profile that was acceptable to move forward into late-phase efficacy studies.

### What did the researchers do and find?

- Participants ($n = 1,288$) in 2 age groups (aged 18 to 59 and 60 to 84 years) were randomly assigned to receive either 1 or 2 intramuscular doses of 5-μg or 25-μg NVX-CoV2373 or placebo.

- Reactogenicity was predominantly mild to moderate in severity and of short duration (median <3 days) after first and second vaccination with NVX-CoV2373, with higher frequencies and intensity after second vaccination and with the higher dose. Reactogenicity occurred less frequently and was of lower intensity in older participants.

- The 2-dose regimen of 5-μg NVX-CoV2373 was found to be immunogenic and well tolerated in both younger and older adults.

### What do these findings mean?

- These findings support the use of the lower-dose regimen in late-phase efficacy studies.

## Introduction

The coronavirus disease 2019 (COVID-19) pandemic caused by the severe acute respiratory syndrome coronavirus 2 (SARS-CoV-2) has continued to spread rapidly throughout the world, with over 174 million confirmed cases and over 3.7 million deaths globally as of June 11, 2021 [1]. Mutated SARS-CoV-2 variants with enhanced transmissibility emerged and established themselves as clinically dominant in the United Kingdom and South Africa in late 2020 and have been spreading globally. Therefore, there is a growing health and economic need for safe and efficacious vaccines to help prevent the spread of SARS-CoV2 and its variants throughout the world.

NVX-CoV2373 contains Matrix-M1 adjuvant [2] and a recombinant SARS-CoV-2 (rSARS-CoV-2) [3] protein, constructed from the full-length (i.e., including the transmembrane domain), wild-type SARS-CoV-2 spike glycoprotein. Viral attachment to the human angiotensin-converting enzyme 2 (hACE2) receptor of host cells is mediated by SARS-CoV2 spike protein and serves as a target for development of antibodies and vaccines [4,5]. Targeting the SARS-CoV-2 spike protein has been shown to be highly effective in preventing COVID-19 in clinical trials, which has formed the basis for emergency use of several vaccine candidates [6,7].

The phase 1 component of the phase 1 to 2 trial showed that 2-dose regimens of 5-μg and 25-μg rSARS-CoV-2 with 50-μg Matrix-M1 adjuvant (mixed prior to use) in participants 18 to 59 years were well tolerated and immunogenic as seen by increases in immunoglobulin G (IgG) anti-spike antibodies and neutralizing antibodies with an inhibitory concentration of >99% ($MN_{>99\%}$), which exceed levels seen in a panel of COVID-19 convalescent serum samples [3]. Importantly, the Matrix-M1 adjuvant was dose sparing and induced CD4+ effector memory T-cell responses that were biased toward a Th1 phenotype, which may play a role in reducing the theoretical possibility of antibody-dependent enhancement (ADE) of SARS-CoV-2 infection [8].

Here, we report on the phase 2 safety and immunogenicity, evaluating 1- or 2-dose regimens for 2 antigen dose levels. This descriptive study was designed to generate data supporting selection of a preferred dose regimen in both younger and older adults for evaluation in clinical endpoint studies. The phase 2 study does not include any participants from the phase 1 component of the study.

## Methods

### Trial design and oversight

Our phase 2 trial was conducted at 9 sites in Australia and 8 sites in the United States. Eligible participants were men and nonpregnant women 18 to 84 years of age with a body mass index (the weight in kilograms divided by the square of the height in meters) of 17 to 35. Participants with a range of underlying medical conditions could be enrolled if the conditions were judged clinically to be stable, and, therefore, the study population was inclusive of potentially more comorbid conditions than would normally be enrolled in a phase 2 study to allow a more generalizable population to be enrolled for expected priority population groups for vaccination. Details of the trial design, eligibility criteria, conduct, oversight, and analyses are provided in the protocol and statistical analysis plan (included in the Supporting information). All participants provided written informed consent prior to trial enrollment.

Up to 750 participants were planned to be randomized in each country (150 participants per treatment group) to decrease the risk that study recruitment or implementation in either country might be difficult due to outbreak restrictions or vaccine and sample transportation

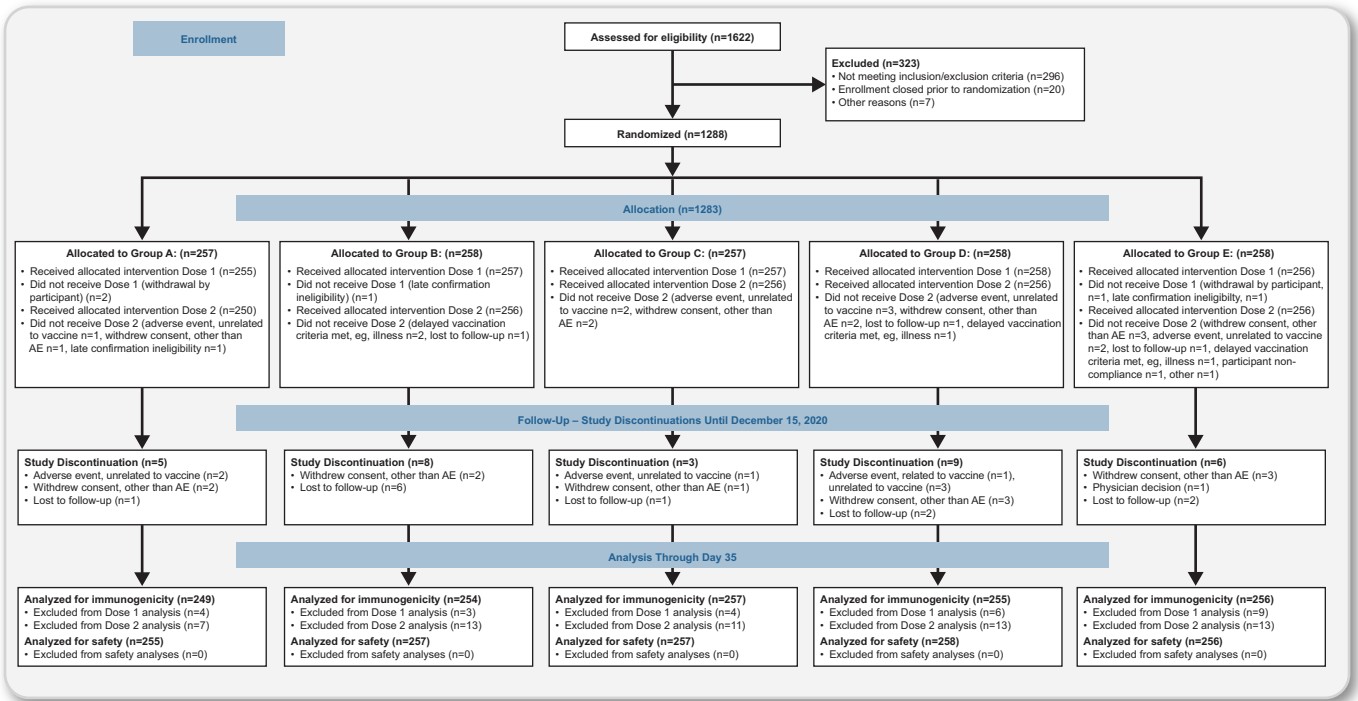

**Fig 1. CONSORT flow diagram.** Notes: Study treatments Dose 1 / Dose 2: Group A: placebo / placebo; Group B: 5-µg SARS-CoV-2 rS + M1 / 5-µg SARS-CoV-2 rS + M1; Group C: 5-µg SARS-CoV-2 rS + M1 / placebo; Group D: 25-µg SARS-CoV-2 rS + M1 / 25-µg SARS-CoV-2 rS + M1; Group E: 25-µg SARS-CoV-2 rS + M1 / placebo (M1 = 50-µg Matrix M1 adjuvant). Details of the analysis sets are included in the Supporting methods section.

challenges. Eligible participants were randomly assigned (1:1:1:1:1) in a blinded manner to 1 of 4 vaccine groups (Groups B, C, D, and E) or placebo (Group A) according to pregenerated randomization schedules and were stratified by age (ages 18 to 59 and 60 to 84 years) and study site (Fig 1). As a safety measure, enrollment of older participants was initially limited to approximately 50 participants in each of the 4 vaccine and placebo groups until reactogenicity was observed for 5 days after which safety monitoring committee review confirmed further enrollment could continue.

Each participant received 2 intramuscular injections of 5-µg (low-dose) or 25-µg (high-dose) NVX-CoV2373 and/or 0.9% sodium chloride placebo in a 1-dose (5 µg or 25 µg followed by placebo) or 2-dose (5 µg followed by 5 µg or 25 µg followed by 25 µg) regimen, 21 days apart. Both dose regimens included 50 µg of the Matrix-M1 adjuvant. Participants and trial site personnel managing the conduct of the trial remained blinded to vaccine assignment. Vaccination pause rules were in place to monitor participants' safety (see Supporting appendix).

The trial was designed by Novavax (Gaithersburg, Maryland, USA), with funding support from the Coalition for Epidemic Preparedness Innovations. The trial protocol was approved by the Alfred Hospital Human Research Ethics Committee (Melbourne, Australia) and Advarra Central Institutional Review Board (Columbia, Maryland, USA) and was performed in accordance with the International Conference on Harmonisation, Good Clinical Practice guidelines; ClinicalTrials.gov number: NCT04368988. Safety oversight for the older adult lead-in enrollment cohort and specific vaccination pause rules were supported by an independent safety monitoring committee. The authors assume responsibility for the accuracy and completeness of the data and analyses, as well as for the fidelity of the trial.

## Trial procedures: Safety assessments

Reactogenicity was self-reported by participants using an electronic smartphone-based diary with daily electronic prompts for completion each evening, starting on the day of vaccination (Day 0) and for 6 subsequent days (for a total of 7 days). Predefined solicited local (injection site) reactogenicity included pain, tenderness, erythema, and swelling; systemic reactogenicity included fever, nausea or vomiting, headache, fatigue, malaise, myalgia, and arthralgia. Reactogenicity severity was graded according to the Food and Drug Administration (FDA) toxicity grading scale with minor modifications (Table A in S1 Text) [9]. Unsolicited adverse events (AEs) were assessed at each study visit and coded by preferred term and system organ class according to the *Medical Dictionary for Regulatory Activities* (MedDRA), version 23.0, and summarized by trial vaccine, clinical severity (mild, moderate, and severe), and causality (not related and related). Definitions for AEs of special interest relevant to COVID-19 [10,11] (Table B in S1 Text) and potential immune-mediated medical conditions (PIMMCs) (Table C in S1 Text) are included in the Supporting information.

## Immunogenicity assessments

An enzyme-linked immunosorbent assay (ELISA) (Novavax, Gaithersburg, Maryland, USA) was used to measure IgG anti-spike protein levels specific for rSARS-CoV-2 protein antigens from all participants with evaluable samples at Days 0 (baseline), 21, and 35, with seropositivity defined as a titer above the lower limit of quantification (LLOQ; 200 ELISA units per milliliter). A microneutralization (MN) assay (360biolabs, Melbourne, Australia) with an inhibitory concentration of 50% ($MN_{50\%}$) was used to measure neutralizing antibodies specific to wild-type virus SARS-CoV-2 at the same time points from a nonrandomized subgroup of approximately 250 participants (25 participants per treatment group per age group) with seropositivity defined as titer above the LLOQ (titer of 20). Although nonrandomized, the MN subset was selected (by an unblinded study statistician) to be approximately balanced by treatment group and age group from participants enrolled earlier in the study until the subgroup quota was met. Details of assays are included in the Supporting methods section. An ad hoc analysis was conducted, comparing the neutralizing antibody response to a sample serum panel from a clinical case series of 60 convalescent sera samples from cases with polymerase chain reaction (PCR)-confirmed asymptomatic to severe symptoms of COVID-19 from Baylor College of Medicine (Houston, Texas, USA). Three samples were from pediatric cases, 13 (22%) of samples were from adults 18 to 49 years, 13 (22%) from adults 50 to 64 years, and 31 (52%) from older adults 65 to 79 years. In terms of clinical disease severity, 11 (18%) were from asymptomatic/mild cases (identified through contact/exposure assessment), 30 (50%) were from mild/moderate cases (outpatient cases/discharged cases from the emergency department), and 19 (32%) were from moderate/severe cases (hospitalized cases/intensive care unit cases).

## Statistical analysis

The decision on which dosing regimen of NVX-CoV2373 to move forward into late-phase studies in both younger and older adult participants was based on the totality of the immunogenicity and safety data rather than any individual measurement and was not based on formal statistical comparisons of treatments. With a minimum of approximately 150 participants in each treatment group (in either country), there was a greater than 99.9% probability of observing at least 1 participant with an AE if the true incidence of the AE was 5% and a 77.9% probability if the true incidence of the AE was 1%. For immunogenicity endpoints, power estimates based on a range of assumptions for potential immunogenicity endpoint scenarios were made and are included in the statistical analysis plan. Most endpoints were summarized using

descriptive statistics, with 95% confidence intervals (CIs) added based on the t-distribution of the log-transformed values. This Set included all participants who received at least 1 dose of study vaccine or placebo. Immunogenicity Analysis Sets are described in the protocol and statistical analysis plan. Any participant who tested SARS-CoV-2 positive by qualitative PCR testing from screening and prior to the immunogenicity assessment for a particular time point were excluded from the Per-Protocol Analysis Set for that time point as were participants who had major protocol deviations that might affect their immune responses.

## Results

### Trial population

Between August 24, 2020, and September 25, 2020, a total of 1,288 participants underwent randomization at 17 sites in Australia and the US in the phase 2 component of the trial (Fig A in S1 Text). A total of 1,283 participants received the initial injection, with 255 in the placebo group (Group A), 258 in the 5-μg 2-dose regimen (Group B), 256 in the 5-μg 1-dose regimen (Group C), 259 in the 25-μg 2-dose regimen (Group D), and 255 in the 25-μg 1-dose regimen (Group E). A total of 1,256 participants received both injections (250, 254, 255, 251, and 246 in Groups A, B, C, D, and E, respectively). Demographic characteristics of the participants based on the Safety Analysis Set population are presented in Table 1. The median age was 57 years, with 45% in the older age group. Among the 1,283 participants, 49% were male, 87% were white, 4% were of Hispanic or Latino origin, and 2% had a positive baseline SARS-CoV-2 serostatus.

### Safety outcomes

An overall summary by treatment group for solicited and unsolicited AEs is included in Table D in S1 Text. The safety results reported are for safety follow-up until the data cutoff date for the Day 35 interim analysis of December 9, 2020, which was inclusive of at least 35 days of follow-up for all enrolled participants. Safety follow-up is ongoing for the study.

### Solicited local reactogenicity

Across both age groups (18 to 59 and 60 to 84 years), solicited local AEs were more common in the NVX-CoV2373 groups than in the placebo group following each vaccination (Fig 2A and Tables E, F, and G in S1 Text).

Following first vaccination of low-dose 5-μg NVX-CoV2373 (Groups B+C) and high-dose 25-μg NVX-CoV2373 (Groups D+E) for all participants 18 to 84 years (Fig 2 and Table E in S1 Text), the most frequent solicited local AEs for "any grade" severity were tenderness (48%, 95% CI 43.6, 52.5 and 59%, 95% CI 55.0, 63.7 for Groups B+C and D+E, respectively) and pain (27%, 95% CI 23.5, 31.5 and 37%, 95% CI 33.0, 41.6, respectively).

Following the first vaccination with low-dose 5-μg NVX-CoV2373 (Group B+C), the frequencies of "any grade" severity solicited systemic events were higher among younger participants for some measures such as tenderness (60.6%, 95% CI 54.6, 66.4) compared with older participants (32.9%, 95% CI 26.9, 39.4). Pain in younger participants was 35.7% (95% CI 30.1, 41.7) and 17.3% (95% CI 12.7, 22.8) in older participants (Fig 2C and Tables F and G in S1 Text). Tenderness and pain were predominantly grade 1 to 2 in younger participants and grade 1 in older participants and of short duration (median of 1 day for pain and 2 days for tenderness) across both age groups. Grade 3 solicited local AEs were reported in 1 low-dose (<1%) and 2 high-dose recipients (1%) among younger participants and in 1 high-dose recipient (<1%) among older participants; there were no grade 4 events.

**Table 1. Demographic characteristics of the participants at enrollment (Safety Analysis Set).**

| AGEU Group | A | B | C | D | E | TOTAL 1,283 |
|---|---|---|---|---|---|---|
| Study Treatments Dose 1 / Dose 2 | Placebo / Placebo | 5 µg + M1 / 5 µg + M1 | 5 µg + M1 / Placebo | 25 µg + M1 / 25 µg + M1 | 25 µg + M1 / Placebo | |
| **Characteristic N** | **255** | **258** | **256** | **259** | **255** | |
| Sex–no. (%) | | | | | | |
| Male | 132 (51.8) | 119 (46.1) | 136 (53.1) | 122 (47.1) | 121 (47.5) | 630 (49.1) |
| Female | 123 (48.2) | 139 (53.9) | 120 (46.9) | 137 (52.9) | 134 (52.5) | 653 (50.9) |
| Age–years | | | | | | |
| All– 18–84 mean (SD) | 51.8 (17.17) | 51.3 (17.47) | 52.7 (16.82) | 52.5 (16.60) | 53.9 (16.14) | 52.4 (16.84) |
| 18–59 no. (% of total) | 139 (54.5) | 140 (54.3) | 139 (54.3) | 144 (55.6) | 138 (54.1) | 700 (54.6) |
| mean (SD) | 38.9 (12.40) | 38.2 (12.54) | 40.2 (12.17) | 40.8 (12.60) | 42.3 (12.45) | 40.1 (12.48) |
| 60–84 no. (% of total) | 116 (45.5) | 118 (45.7) | 117 (45.7) | 115 (44.4) | 117 (45.9) | 583 (45.4) |
| mean (SD) | 67.1 (5.31) | 66.9 (5.71) | 67.5 (5.94) | 67.2 (5.55) | 67.7 (5.90) | 67.3 (5.68) |
| 75–84 no. (% of total) | 13 (5.1) | 14 (5.4) | 18 (7.0) | 17 (6.6) | 21 (8.2) | 83 (6.5) |
| mean (SD) | 77.1 (2.53) | 77.5 (2.38) | 78.3 (2.33) | 76.8 (1.56) | 77.8 (3.06) | 77.5 (2.46) |
| Race or ethnic group–no. (%) | | | | | | |
| White | 228 (89.4) | 225 (87.2) | 216 (84.4) | 222 (85.7) | 223 (87.5) | 1,114 (86.8) |
| Black or African American | 6 (2.4) | 7 (2.7) | 7 (2.7) | 11 (4.2) | 6 (2.4) | 37 (2.9) |
| Asian | 16 (6.3) | 18 (7.0) | 25 (9.8) | 20 (7.7) | 19 (7.5) | 98 (7.6) |
| American Indian or Alaska Native | 2 (0.8) | 2 (0.8) | 2 (0.8) | 1 (0.4) | 1 (0.4) | 8 (0.6) |
| Native Hawaiian or other Pacific Islander | 0 | 0 | 0 | 0 | 1 (0.4) | 1 (<0.1) |
| Multiracial | 3 (1.2) | 3 (1.2) | 3 (1.2) | 3 (1.2) | 3 (1.2) | 15 (1.2) |
| Not reported/missing | 0 | 3 (1.2) | 3 (1.2) | 2 (0.8) | 2 (0.8) | 10 (0.8) |
| Hispanic or Latino | 15 (5.9) | 6 (2.3) | 12 (4.7) | 7 (2.7) | 11 (4.3) | 51 (4.0) |
| Body mass index*[†] | 26.76 (4.244) | 27.10 (4.121) | 26.35 (4.232) | 27.26 (3.930) | 26.78 (4.067) | 26.85 (4.126) |
| Baseline SARS-CoV-2 Serostatus–no. (%) | | | | | | |
| Seropositive[‡] | 6 (2.4) | 5 (1.9) | 6 (2.3) | 7 (2.7) | 7 (2.7) | 31 (2.4) |
| Country of participants–no. (%) | | | | | | |
| Australia | 130 (51.0) | 134 (51.9) | 132 (51.6) | 133 (51.4) | 130 (51.0) | 659 (51.4) |
| United States | 125 (49.0) | 124 (48.1) | 124 (48.4) | 126 (48.6) | 125 (49.0) | 624 (48.6) |

*Means ± SD.

[†]The body mass index is the weight in kilograms divided by the square of the height in meters. This calculation was based on the weight and height measured at the time of screening.

[‡]Seropositive—Day 0 seropositivity to serum IgG antibody to SARS-CoV-2 proteins.

IgG, immunoglobulin G; SARS-CoV-2, severe acute respiratory syndrome coronavirus 2; SD, standard deviation.

Following second vaccination of 5-µg (Group B) and 25-µg (Group D) NVX-CoV2373, the most frequent solicited local AEs for all participants 18 to 84 years (Fig 2A and Table E in S1 Text) were tenderness (Group B: 65.2%, 95% CI 58.9, 71.1 and Group D: 76.1, 95% CI 70.3, 81.3) and pain (Group B: 45.6%, 95% CI 39.3, 52.0 and Group D: 54.7%, 95% CI 48.2, 61.0). For the 2-dose regimen of low-dose 5-µg NVX-CoV2373 (Group B), the frequencies of "any grade" severity solicited systemic events were higher among younger participants for some measures such as tenderness (73.7%, 95% CI 65.5, 80.9) compared with older participants (54.9%, 95% CI 45.2, 64.2). Pain in younger participants was 49.6% (95% CI 41.0, 58.3) and 40.7% (95% CI 31.6, 50.4) in older participants (Fig 2C and Tables F and G in S1 Text).

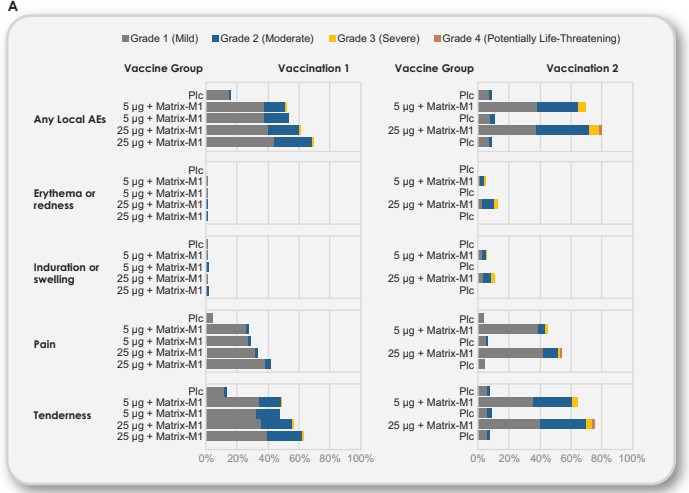

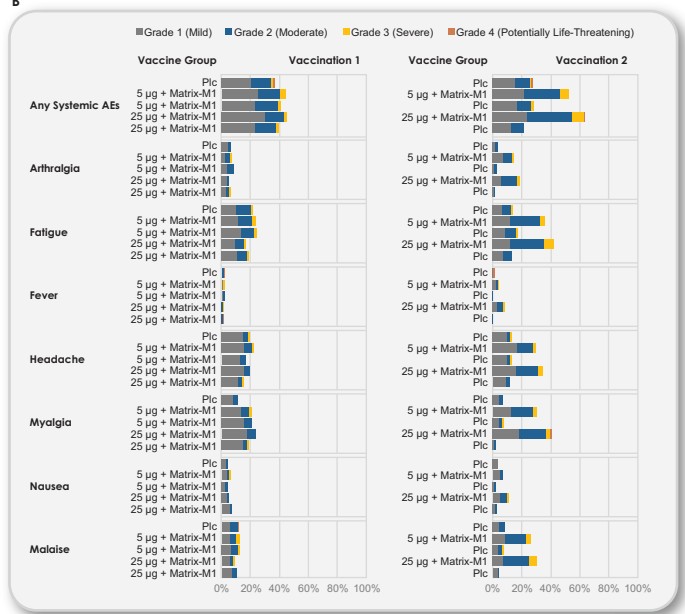

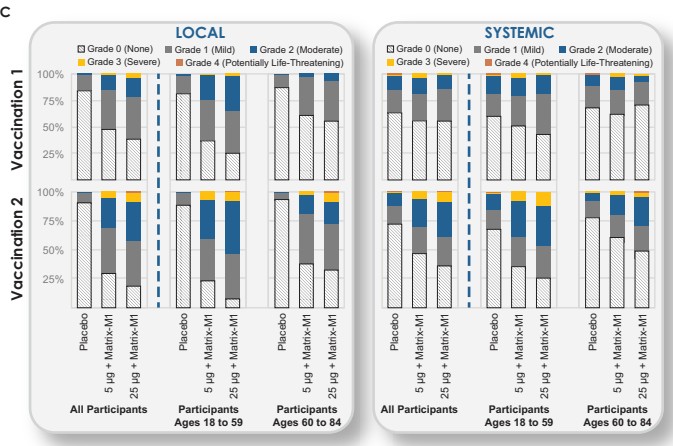

**Fig 2. Solicited local and systemic AEs.** The percentage of participants in each vaccine group (Groups A, B, C, D, and E) with solicited local and systemic AEs according to the maximum toxicity grade (mild, moderate, severe, or potentially life-threatening) during the 7 days after each vaccination is plotted for solicited local (Panel A) and systemic (Panel B) AEs and for both local and systemic events by age group (Panel C). Grade 4 (potentially life-threatening) events met these criteria by emergency department attendances in Australia, where nonemergency conditions are frequently managed as an alternative to regular primary care. (See Tables E, F, and G in S1 Text for complete solicited local and systemic AE data by dose, by all participants, and by younger and older age groups.) (**A**) Solicited local AEs. (**B**) Solicited systemic AEs. (**C**) Reactogenicity by age. AE, adverse event; Plc, placebo.

Tenderness and pain were predominantly grade 1 to 2 in younger participants and grade 1 in older participants and of short duration (median of 2 days) across both age groups.

For vaccine participants who received placebo as the second injection following active vaccine at Dose 1 (Groups C and E), frequencies of solicited local AEs were similar to placebo (Group A) after the second vaccination. Grade 3 solicited local AEs were reported in 10 low-dose recipients (7%) and 10 high-dose recipients (7%) among younger participants and in 3 low-dose NVX-CoV2373 recipients (3%) and 9 high-dose NVX-CoV2373 recipients (8%) among older participants; grade 4 events were reported in 1 high-dose recipient (1%) in each age group.

## Solicited systemic reactogenicity

Across both age groups, solicited systemic AEs other than muscle pain were reported at similar frequencies between the NVX-CoV2373 and placebo groups following first vaccination; however, solicited systemic AEs were more common in participants administered a second dose of the active NVX-CoV2373 vaccine (Groups B and D) than in the other groups following second vaccination (Fig 2B and Tables E, F, and G in S1 Text). Fever was infrequently reported in 1% to 2% of vaccine recipients across both age groups, with grade 3 fever reported in 3 low-dose recipients (1%) and 1 high-dose recipient (<1%) (Table E in S1 Text).

For younger participants 18 to 64 years following first vaccination, Grade 3 solicited systemic AEs were infrequent, being reported in 2 placebo recipients (1%), 7 low-dose recipients (3%), and 4 high-dose recipients (1%), with 1 grade 4 event (fever) reported in 1 placebo recipient (1%).

Among older participants 65 to 84 years following first vaccination, grade 3 solicited systemic AEs were reported in 6 low-dose recipients (3%) and 2 high-dose recipients (1%), and 1 grade 4 event (malaise) was reported in 1 placebo recipient (1%). The grade 4 malaise was associated with an emergency department attendance for a participant with concurrent atrial fibrillation who was not subsequently admitted to hospital.

Following second vaccination of 5-μg (Group B) and 25-μg (Group D) NVX-CoV2373, the most frequent solicited systemic AEs were fatigue (Group B: 35.6%, 95% CI 29.7, 41.9 and Group D: 42.5%, 95% CI 36.3, 48.9) and muscle pain (Group B: 30.8%, 95% CI 25.1, 36.9 and Group D: 40.9%, 95% CI 34.7, 47.3).

For the 2-dose regimen of low-dose 5-μg NVX-CoV2373 (Group B), the frequencies of any solicited systemic events were higher among younger participants (64.2%, 95% CI 55.6, 72.2) compared with older participants (38.9%, 95% CI 29.9, 48.6). This was also seen for the more frequent measures such as fatigue–younger participants 46.0% (95% CI 37.4, 54.7) compared with older participants 23.0% (95% CI 15.6, 31.9) and muscle pain–younger participants 40.1% (95% CI 31.9, 48.9) compared with older participants 19.5% (95% CI 12.6, 28.0). Systemic AEs were generally grade 1 or 2 in severity and of short duration (median of 1 day) across both age groups.

Following second vaccination for younger participants, "any grade" fever was reported in 1 placebo recipient (1%), 9 low-dose recipients (7%), and 16 high-dose recipients (12%), with

grade 3 fever reported in 1 low-dose recipient (1%) and 2 high-dose recipients; grade 4 fever was reported in 1 placebo recipient (1%). Among older participants following second vaccination, "any grade" fever was reported in 1 placebo recipient (1%), 2 low-dose recipients (2%), and 4 high-dose recipients (4%), with grade 3 fever reported in 1 high-dose recipient (1%); no grade 4 fever events were reported.

Grade 3 solicited systemic AEs in younger participants following the second vaccination were reported in 1 placebo recipient (1%), 11 low-dose recipients (8%), and 16 high-dose recipients (12%), and 1 grade 4 event was reported in a placebo recipient (<1%).

Among older participants, grade 3 solicited systemic AEs following the second vaccination were reported in 1 placebo recipient (1%), 3 low-dose recipients (3%), and 4 high-dose recipients (4%). A grade 4 event of muscle pain was reported in 1 high-dose recipient (1%) assessed for an injection site reaction at an emergency department, which did not require hospital admission and subsequently resolved. No solicited grade 4 AEs after either vaccination dose met seriousness criteria for serious adverse events (SAEs).

### Unsolicited adverse events

Unsolicited AEs were reported (Table H in S1 Text) in 42 placebo recipients (17%), 86 low-dose recipients (17%), and 95 high-dose recipients (18%), with similar distributions across both age groups. Unsolicited AEs were predominantly mild across both age groups, with severe events reported in 3 placebo recipients (1%), 5 low-dose recipients (1%), and 1 high-dose recipient (<1%). One low-dose recipient (<1%) had a related severe unsolicited AE (acute colitis) that also met the criteria for an SAE.

Seven participants discontinued the trial due to an unsolicited AE, including 2 placebo recipients (non-Hodgkin lymphoma; atrial fibrillation); 1 low-dose recipient (urinary incontinence); and 4 high-dose recipients (arthralgia; pyrexia, myalgia, and malaise (related); dermatitis; and atrial fibrillation).

Nine SAEs were reported with 2 SAEs assessed as related by the investigator; 1 case of acute colitis in a low-dose recipient, and 1 case of multiple sclerosis in a placebo recipient (which also met criteria for a PIMMC). The other 7 SAEs reported as unrelated by investigators were single cases of atrial fibrillation (assessed as not related by the investigator due to underlying cardiac disease), myocardial infarction (assessed as not related due to underlying risk factors of hypertension, type 2 diabetes, and hypercholesterolemia), vertigo, wrist fracture, non-Hodgkin lymphoma, animal bite, and lumbar spinal stenosis. There were no AEs of special interest associated with COVID-19 as defined by regulatory criteria for conditions listed in Table S2.

### Immunogenicity outcomes

**Anti-spike protein binding IgG response.** For the primary endpoint analysis at Day 35 (14 days following the second dose), participants 18 to 84 years of age who received full vaccination regardless of baseline serostatus had high and similar anti-spike protein binding IgG geometric mean titers (GMTs) for either of the 2-dose regimens of 5-μg and 25-μg NVX-CoV2373 (44,421 and 46,459, respectively; Table 2).

These responses equated to large geometric mean fold rises (GMFRs) of 386 and 385, respectively, relative to baseline. Seroconversion rates (defined as the percentage of participants with a postvaccination titer increase of ≥4-fold) were 98% and 100%, respectively, compared with 1.3% for placebo (Table 2).

Across the 2 age groups, 97.3% of participants were seronegative at baseline with anti-spike protein binding IgG GMTs generally below the limit of quantification (200) at Day 0 (Table 2), reflecting the low rate of previous virus exposure in this population. Following first vaccination

**Table 2. Anti-S protein serum IgG antibody immune responses for NVX-CoV2373 study treatments by study group and age (18 to 84 years), or age subgroup (18 to 59 years and 60 to 84 years).**

| Group | Group A | Group B | Group C | Group D | Group E |
|---|---|---|---|---|---|
| Study Treatments Dose 1 / Dose 2 | Placebo / Placebo | 5 µg + M1 / 5 µg + M1 | 5 µg + M1 / Placebo | 25 µg + M1 / 25 µg + M1 | 25 µg + M1 / Placebo |
| **Primary Endpoint Analysis—Participants 18 to 84 Years of Age (PP Immunogenicity Analysis Set)** | | | | | |
| Day 35 | | | | | |
| n1 | 238 | 240 | 241 | 236 | 243 |
| GMT (EU/mL) (95% CI) | 126.1 (114.0, 139.4) | 44,420.9 (37,929.1, 52,023.8) | 894.0 (744.1, 1,074.0) | 46,459.3 (40,839.4, 52,852.5) | 1,951.3 (1,658.3, 2,296.1) |
| GMFR referencing Day 0 (95% CI) | 1.0 (1.0, 1.1) | 385.6 (325.5, 456.8) | 7.4 (6.3, 8.7) | 384.9 (334.7, 442.7) | 15.4 (13.3, 17.9) |
| SCR ≥ 4-fold increase, n2/n1 (%) (95% CI) | 3/238 (1.3) (0.3, 3.6) | 236/240 (98.3) (95.8, 99.5) | 163/241 (67.6) (61.3, 73.5) | 235/236 (99.6) (97.7, 100.0) | 211/243 (86.8) (81.9, 90.8) |
| **Secondary Endpoint Analyses—Participants 18 to 59 Years of Age (PP Analysis Set)** | | | | | |
| Day 0 | | | | | |
| n1 | 138 | 137 | 140 | 143 | 139 |
| GMT (EU/mL) (95% CI) | 116.5 (105.2, 128.9) | 119.1 (105.6, 134.2) | 123.4 (111.3, 136.8) | 125.2 (110.7, 141.7) | 119.6 (107.6, 132.9) |
| Day 21 | | | | | |
| n1 | 137 | 135 | 139 | 142 | 133 |
| GMT (EU/mL) (95% CI) | 121.0 (108.7, 134.7) | 1,374.0 (1,109.4, 1,701.7) | 1,457.8 (1,170.6, 1,815.4) | 3,155.7 (2,543.3, 3,915.7) | 2,897.6 (2,341.4, 3,586.0) |
| GMFR referencing Day 0 (95% CI) | 1.0 (1.0, 1.1) | 11.5 (9.4, 14.1) | 11.8 (9.6, 14.5) | 25.2 (20.5, 30.9) | 24.0 (19.7, 29.3) |
| SCR ≥ 4-fold increase, n2/n1 (%) (95% CI) | 2/137 (1.5) (0.2, 5.2) | 106/135 (78.5) (70.6, 85.1) | 110/139 (79.1) (71.4, 85.6) | 133/142 (93.7) (88.3, 97.1) | 123/133 (92.5) (86.6, 96.3) |
| Day 35 | | | | | |
| n1 | 135 | 127 | 134 | 137 | 133 |
| GMT (EU/mL) (95% CI) | 123.9 (109.0, 140.9) | 65,019.1 (55,484.8, 76,191.9) | 1,493.4 (1,206.1, 1,849.2) | 58,773.8 (51,611.7, 66,929.8) | 2,644.3 (2,175.6, 3,213.9) |
| GMFR referencing Day 0 (95% CI) | 1.1 (1.0, 1.2) | 538.6 (442.1, 656.2) | 12.1 (9.9, 14.9) | 464.7 (395.2, 546.4) | 21.9 (18.3, 26.3) |
| SCR ≥ 4-fold increase, n2/n1 (%) (95% CI) | 2/135 (1.5) (0.2, 5.2) | 126/127 (99.2) (95.7, 100.0) | 109/134 (81.3) (73.7, 87.5) | 137/137 (100.0) (97.3, 100.0) | 125/133 (94.0) (88.5, 97.4) |
| **Secondary Endpoint Analyses—Participants 60 to 84 Years of Age (PP Analysis Set)** | | | | | |
| Day 0 | | | | | |
| n1 | 111 | 117 | 117 | 112 | 117 |
| GMT (MN50) (95% CI) | 125.9 (109.6, 144.6) | 113.0 (102.9, 124.0) | 117.6 (103.6, 133.5) | 115.5 (106.0, 125.8) | 134.0 (116.7, 153.9) |
| Day 21 | | | | | |
| n1 | 108 | 116 | 114 | 107 | 114 |
| GMT (EU/mL) (95% CI) | 120.0 (106.0, 135.9) | 456.4 (361.9, 575.6) | 403.6 (308.8, 527.4) | 746.3 (592.2, 940.5) | 1214.3 (932.7, 1580.8) |
| GMFR referencing Day 0 (95% CI) | 1.0 (0.9, 1.0) | 4.0 (3.3, 5.0) | 3.5 (2.8, 4.3) | 6.5 (5.2, 8.1) | 9.0 (7.1, 11.4) |
| SCR ≥ 4-fold increase, n2/n1 (%) (95% CI) | 0/108 (0.0) (0.0, 3.4) | 53/116 (45.7) (36.4, 55.2) | 46/114 (40.4) (31.3, 49.9) | 73/107 (68.2) (58.5, 76.9) | 92/114 (80.7) (72.3, 87.5) |
| Day 35 | | | | | |
| n1 | 107 | 114 | 112 | 105 | 110 |
| GMT (EU/mL) (95% CI) | 127.8 (109.2, 149.6) | 28,136.6 (21,616.6, 36,623.3) | 457.9 (351.1, 597.1) | 32,871.2 (26,189.5, 41,257.5) | 1,347.5 (1,041.5, 1,743.5) |
| GMFR referencing Day 0 (95% CI) | 1.0 (0.9, 1.1) | 257.7 (197.1, 336.9) | 3.9 (3.1, 4.9) | 286.3 (226.6, 361.8) | 9.9 (7.9, 12.3) |

*(Continued)*

**Table 2.** (Continued)

| Group | Group A | Group B | Group C | Group D | Group E |
|---|---|---|---|---|---|
| Study Treatments Dose 1 / Dose 2 | Placebo / Placebo | 5 µg + M1 / 5 µg + M1 | 5 µg + M1 / Placebo | 25 µg + M1 / 25 µg + M1 | 25 µg + M1 / Placebo |
| SCR ≥ 4-fold increase, n2/n1 (%), (95% CI) | 1/107 (0.9) (0.0, 5.1) | 111/114 (97.4) (92.5, 99.5) | 55/112 (49.1) (39.5, 58.7) | 104/105 (99.0) (94.8, 100.0) | 86/110 (78.2) (69.3, 85.5) |

CI, confidence interval; COVID-19, coronavirus disease 2019; GMFR, geometric mean fold rise; GMT, geometric mean titer; IgG, immunoglobulin G; LLOQ, lower limit of quantification; M1, Matrix M1 adjuvant 50 µg; $MN_{50}$, microneutralization titer expressed as the reciprocal of the highest dilution at which 50% of the replicate wells are protected from infection; n1, number of participants in the PP Analysis Set within each visit with non-missing data; PP, per-protocol; rSARS-CoV-2, recombinant severe acute respiratory syndrome coronavirus 2 spike protein nanoparticle vaccine; SARS-CoV-2, severe acute respiratory syndrome coronavirus 2; SCR, seroconversion rate.

$MN_{50}$ titer LLOQ = 20. Percentages were calculated as (n2/n1) × 100. Titer values less than LLOQ were replaced by 0.5 × LLOQ. The 95th percentile was calculated relative to placebo participants who remain COVID-19 free at the applicable visit. The 95% CI for GMT and GMFR were calculated based on the t-distribution of the log-transformed values, then back transformed to the original scale for presentation. The 95% CI for SCR was calculated using the exact Clopper–Pearson method. 95% CI for the difference of SCR was calculated using the method of Miettinen and Nurminen [12].

(Day 21) with 5-µg NVX-CoV2373 (Groups B and C combined) and 25-µg NVX-CoV2373 (Groups D and E combined), anti-spike protein binding IgG GMTs were higher for younger participants than older participants (Table 2). Seroconversion rates were also higher for younger participants (79% and 93% for 5-µg first doses and 25-µg first doses, respectively, and 43% and 75% for older participants by dose, respectively).

For participants administered active vaccine for both doses (Group B—2 doses of 5-µg NVX-CoV2373 and Group D—2 doses of 25-µg NVX-CoV2373), there was a large increase in anti-spike protein binding IgG GMTs at Day 35 following the second dose relative to post-first-dose immune responses, and, again, responses were higher in younger participants relative to older participants. Anti-spike IgG immune responses were similar for Groups B and D. For Group B participants (2 doses of 5-µg NVX-CoV2373) at Day 35, anti-spike protein binding IgG GMTs were 65,019, the GMFR was 539, and the seroconversion rate was 99% for younger participants (Table 2). For older participants, anti-spike protein binding IgG GMTs were 28,137 the GMFR was 258, and the seroconversion rate was 97% (Table 2).

## Neutralizing antibody responses

A similar pattern of immune responses following study treatments and dose time point for age subgroups were seen for the neutralizing antibody responses in the subset of participants for which neutralizing antibody responses were measured (Tables 2 and 3). At baseline and for the Day 35 time point (following second vaccination), approximately 25 participants per study group and per age subgroup had evaluable results (Table 3). A smaller subset of participants had neutralizing antibody responses evaluated after the first dose (Day 21). The mean age and sex split of participants in the MN subset (with evaluable baseline MN values) were similar to that see in the general study demographics.

At baseline, neutralizing antibodies were below the LLOQ of the assay for most participants, indicating infrequent prior SARS-CoV-2 infection (Table 3).

Neutralizing antibody responses by GMT ($MN_{50}$), GMFR, and seroconversion rate rose following first-dose vaccination for all nonplacebo study groups, with higher values for younger participants relative to older participants (Table 3).

Neutralizing antibody responses were similar for Groups B and D, the 2-dose active vaccination regimens. For Group B participants (2 doses of 5-µg NVX-CoV2373) at Day 35, neutralizing antibody GMTs ($MN_{50}$) were 2,201, the GMFR was 220, and the seroconversion rate

**Table 3. Microneutralization immune responses ($MN_{50}$) for NVX-CoV2373 study treatments by study group and age subgroup (18 to 59 years and 60 to 84 years).**

| Group | Group A | Group B | Group C | Group D | Group E |
|---|---|---|---|---|---|
| Study Treatments Dose 1 / Dose 2 | Placebo / Placebo | 5 µg + M1 / 5 µg + M1 | 5 µg + M1 / Placebo | 25 µg + M1 / 25 µg + M1 | 25 µg + M1 / Placebo |
| **Secondary Endpoint Analyses—Participants 18 to 59 Years of Age (PP Analysis Set)** | | | | | |
| **Day 0** | | | | | |
| n1 | 26 | 24 | 31 | 23 | 24 |
| GMT ($MN_{50}$) (95% CI) | 10.0 (10.0, 10.0) | 10.0 (10.0, 10.0) | 10.0 (10.0, 10.0) | 10.0 (10.0, 10.0) | 10.0 (10.0, 10.0) |
| **Day 21** | | | | | |
| n1 | 11 | 8 | 12 | 11 | 10 |
| GMT ($MN_{50}$) (95% CI) | 10.0 (10.0, 10.0) | 36.7 (12.3, 109.4) | 44.9 (26.5, 75.9) | 132.4 (64.2, 273.2) | 60.6 (17.9, 205.2) |
| GMFR referencing Day 0 (95% CI) | 1.0 (1.0, 1.0) | 3.7 (1.2, 10.9) | 4.5 (2.7, 7.6) | 13.2 (6.4, 27.3) | 6.1 (1.8, 20.5) |
| SCR ≥ 4-fold increase, n2/n1 (%) (95% CI) | 0/11 (0.0) (0.0, 28.5) | 3/8 (37.5) (8.5, 75.5) | 8/12 (66.7) (34.9, 90.1) | 10/11 (90.9) (58.7, 99.8) | 7/10 (70.0) (34.8, 93.3) |
| **Day 35** | | | | | |
| n1 | 26 | 23 | 29 | 23 | 23 |
| GMT ($MN_{50}$) (95% CI) | 11.4 (8.7, 15.0) | 2,200.8 (1,342.6, 3,607.5) | 26.6 (18.1, 39.3) | 1,783.1 (1,191.9, 2,667.7) | 52.5 (31.3, 88.0) |
| GMFR referencing Day 0 (95% CI) | 1.1 (0.9, 1.5) | 220.1 (134.3, 360.7) | 2.7 (1.8, 3.9) | 178.3 (119.2, 266.8) | 5.2 (3.1, 8.8) |
| SCR ≥ 4-fold increase, n2/n1 (%)(95% CI) | 1/26 (3.8) (0.1, 19.6) | 23/23 (100.0) (85.2, 100.0) | 13/29 (44.8) (26.4, 64.3) | 23/23 (100.0) (85.2, 100.0) | 15/23 (65.2) (42.7, 83.6) |
| **Secondary Endpoint Analyses—Participants 60 to 84 Years of Age (PP Analysis Set)** | | | | | |
| **Day 0** | | | | | |
| n1 | 25 | 27 | 26 | 26 | 24 |
| GMT ($MN_{50}$) (95% CI) | 10.0 (10.0, 10.0) | 11.1 (9.0, 13.7) | 10.0 (10.0, 10.0) | 10.0 (10.0, 10.0) | 10.0 (10.0, 10.0) |
| **Day 21** | | | | | |
| n1 | 10 | 13 | 10 | 10 | 10 |
| GMT ($MN_{50}$) (95% CI) | 10.0 (10.0, 10.0) | 42.2 (15.7, 113.5) | 18.7 (10.8, 32.2) | 32.5 (14.0, 75.6) | 32.5 (15.5, 68.2) |
| GMFR referencing Day 0 (95% CI) | 1.0 (1.0, 1.0) | 3.4 (1.7, 6.9) | 1.9 (1.1, 3.2) | 3.2 (1.4, 7.6) | 3.2 (1.5, 6.8) |
| SCR ≥ 4-fold increase, n2/n1 (%) (95% CI) | 0/10 (0.0) (0.0, 30.8) | 6/13 (46.2) (19.2, 74.9) | 3/10 (30.0) (6.7, 65.2) | 5/10 (50.0) (18.7, 81.3) | 5/10 (50.0) (18.7, 81.3) |
| **Day 35** | | | | | |
| n1 | 25 | 26 | 26 | 26 | 24 |
| GMT ($MN_{50}$) (95% CI) | 10.0 (10.0, 10.0) | 980.5 (559.8, 1717.1) | 15.7 (12.1, 20.4) | 1,034.2 (639.8, 1,671.6) | 27.5 (17.9, 42.3) |
| GMFR referencing Day 0 (95% CI) | 1.0 (1.0, 1.0) | 98.0 (56.0, 171.7) | 1.6 (1.2, 2.0) | 103.4 (64.0, 167.2) | 2.7 (1.8, 4.2) |
| SCR ≥ 4-fold increase, n2/n1 (%)(95% CI) | 0/25 (0.0) (0.0, 13.7) | 26/26 (100.0) (86.8, 100.0) | 6/26 (23.1) (9.0, 43.6) | 25/26 (96.2) (80.4, 99.9) | 11/24 (45.8) (25.6, 67.2) |

CI, confidence interval; COVID-19, coronavirus disease 2019; GMFR, geometric mean fold rise; GMT, geometric mean titer; LLOQ, lower limit of quantification; $MN_{50}$, microneutralization titer expressed as the reciprocal of the highest dilution at which 50% of the replicate wells are protected from infection; n1, number of participants in the PP Analysis Set within each visit with non-missing data; PP, per-protocol; rSARS-CoV-2, recombinant severe acute respiratory syndrome coronavirus 2 spike protein nanoparticle vaccine; SARS-CoV-2, severe acute respiratory syndrome coronavirus 2; SCR, seroconversion rate.

$MN_{50}$ titer LLOQ = 20. Percentages were calculated as (n2/n1) × 100. Titer values less than LLOQ were replaced by 0.5 × LLOQ. The 95th percentile was calculated relative to placebo participants who remain COVID-19 free at the applicable visit. The 95% CI for GMT and GMFR were calculated based on the t-distribution of the log-transformed values, then back transformed to the original scale for presentation. The 95% CI for SCR was calculated using the exact Clopper–Pearson method. 95% CI for the difference of SCR was calculated using the method of Miettinen and Nurminen [12].

was 100% for younger participants (Table 2). For older participants, neutralizing antibody GMTs ($MN_{50}$) at Day 35 were 981, the GMFR was 98, and the seroconversion rate was 100% (Table 3).

Neutralizing antibody responses of the 2-dose regimens of 5-µg (Group B) and 25-µg (Group D) NVX-CoV2373 in both younger and older participants exceeded those seen in a panel of convalescent sera (Fig 3).

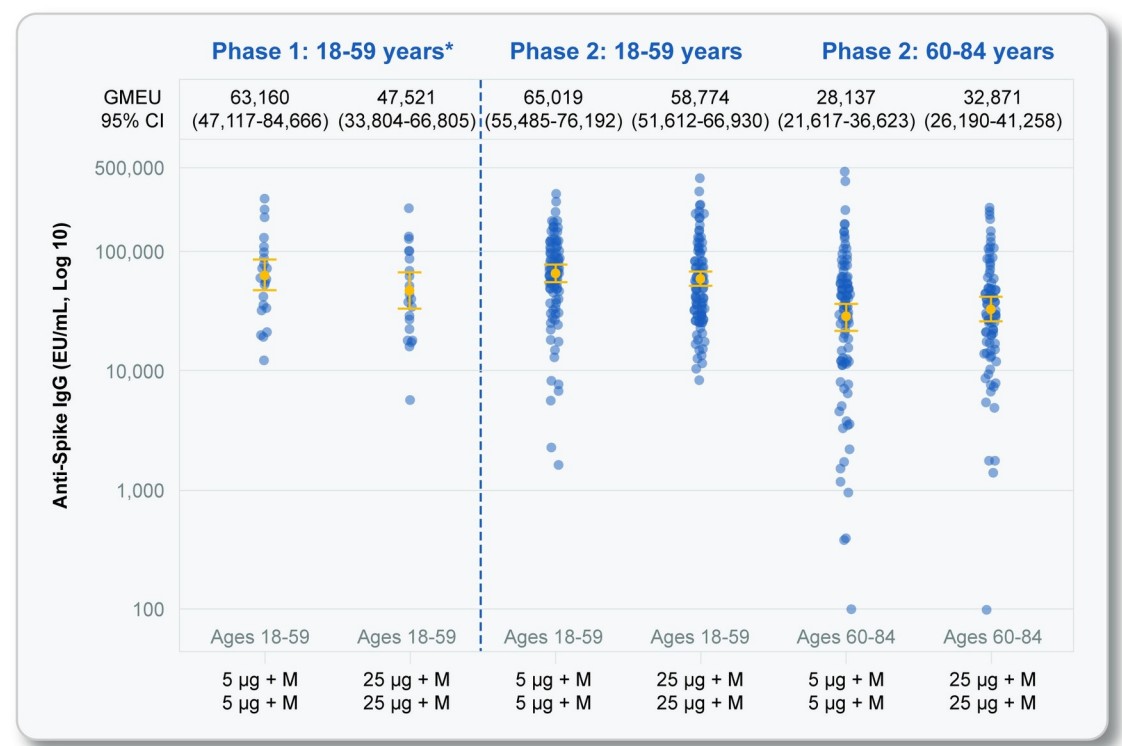

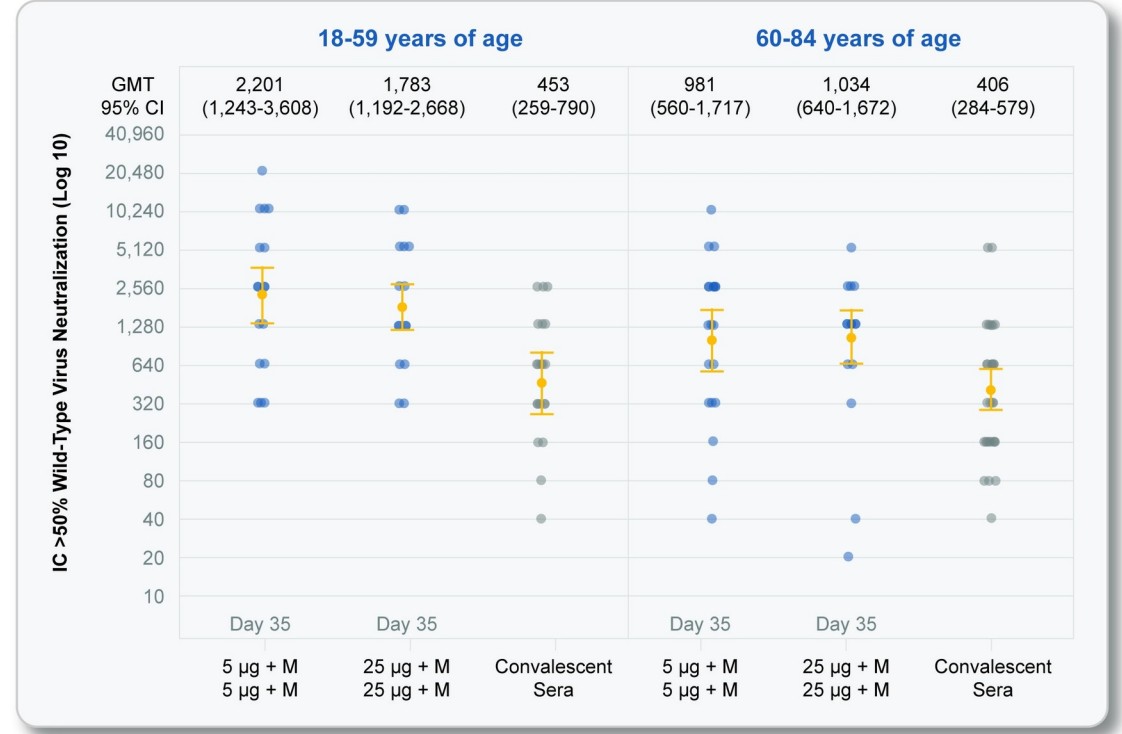

**Fig 3. SARS-CoV-2 anti-spike IgG and neutralizing antibody responses at Day 35.** Shown are geometric mean anti-spike IgG ELISA unit responses to the NVX-CoV2373 protein antigen (Panel A) for younger adult and older adult age groups in the phase 2 study for the 2-dose primary vaccination treatment regimens (Groups B and D), along with aligned group results by the same assay in the phase 1 study, and wild-type SARS-CoV-2 MN assay at an inhibitory concentration >50% ($MN_{50}$) titer responses (Panel B) for younger adult and older adult age groups in the phase 2 study. These panels also include immune responses from a panel of participants by age group from a clinical case series of participants (see Methods). In Panels A and B, boxes and horizontal bars represent IQR and median area under the curve, respectively. Whisker endpoints are equal to the maximum and minimum values below or above the median ± 1.5 times the IQR. (**A**) SARS-CoV-2 anti-spike IgG ELISA. (**B**) Wild-type SARS-CoV-2 MN. CI, confidence interval; ELISA, enzyme-linked immunosorbent assay; EU, endotoxin unit; GMT, geometric mean titer; IC, inhibitory concentration; IgG, immunoglobulin G; IQR, interquartile range; MN, microneutralization; SARS-CoV-2, severe acute respiratory syndrome coronavirus 2.

## Discussion

Based on the totality of the safety and immunogenicity data through at least Day 35 from this phase 2 study, the 2-dose regimen of 5-μg NVX-CoV2373 administered 21 days apart was determined to be the preferred dose regimen for phase 3 development and potential licensure in both younger and older adults. Following first vaccination, both dose levels of NVX-CoV2373 were well tolerated, but there was a higher incidence of local reactogenicity (any solicited local AE) with the higher dose for all participants; however, anti-spike protein binding IgG levels and neutralizing antibody responses were similar by dose level. Following second vaccination, both dose levels of NVX-CoV2373 remained generally well tolerated despite increased frequencies and intensities of local and systemic reactogenicity in both younger and older adults, with higher incidence of overall solicited local and systemic AEs seen with the higher-dose level. Most solicited AEs were of mild or moderate grade severity, and grade 3 or higher solicited AEs were uncommon. NVX-CoV2373 induced robust levels of anti-spike protein binding IgG levels and neutralizing antibodies in both younger and older adults, with seroconversion rates of at least 96% achieved with similar immune responses between the 2 dose levels. Notably, in both age groups, there was a high correlation between "binding" antibodies (anti-spike IgG) and neutralizing antibodies (Fig 4), suggesting that the anti-spike immunity was predominantly functional, as indicated by neutralization. Based on the antibody responses, the low-dose, 2-immunization regimen of 5-μg NVX-CoV2373 was selected to move into later phase development. The study is ongoing, with immune persistence being assessed at approximately 6 months, with subsequent booster doses administered to some study subgroups. Cell-mediated immune responses in a subset of participants by analysis of peripheral blood mononuclear cells (PBMCs) for T-cell responses are also underway and will be published once complete.

The 2-dose regimen of 5-μg NVX-CoV2373 is being investigated in a phase 2a/b study in South Africa [NCT04533399] and 2 large phase 3 studies in the UK [NCT04583995] and the US [NCT04611802]. To date, the phase 2a/b has reported that among 2,684 baseline seronegative participants (94% HIV–negative and 6% HIV–positive), that vaccine efficacy was 49.4%, 95% CI, 6.1 to 72.8. Vaccine efficacy among HIV–negative participants was 60.1% (95% CI, 19.9 to 80.1). This vaccine efficacy is predominantly against the B.1.351 variant, with 38/41 (92.7%) of sequenced samples being the B.1.351 variant [13].

As expected, we found lower frequencies of local and systemic reactogenicity and lower anti-spike protein binding IgG and neutralizing antibody responses in older adults. These findings are consistent with other SARS-CoV-2 vaccines [6,7]. Despite the lower immune response in older adults, seroconversion rates after the 2-dose regimen of 5-μg NVX-CoV2373 were 97% for anti-spike protein binding IgG and 100% for neutralizing antibodies, which reflected increases from baseline of >250-fold and >95-fold, respectively. Additionally, in both younger and older adults, neutralizing antibody titers exceeded those seen in a sample of

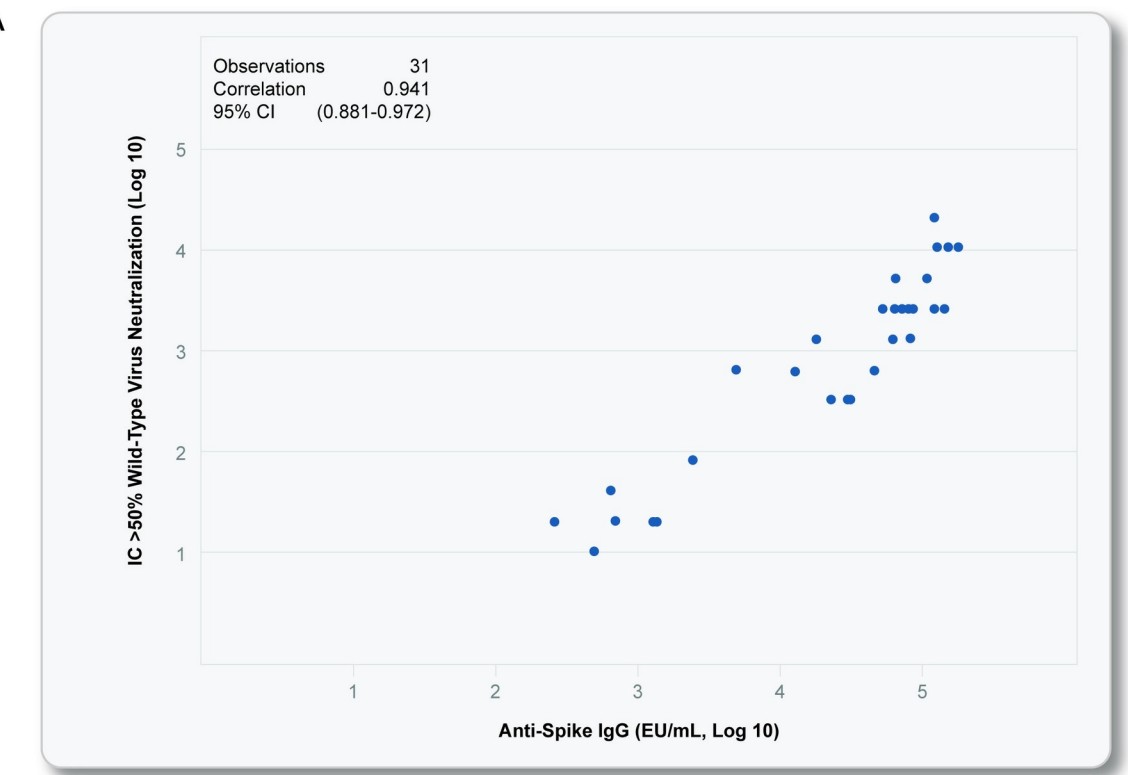

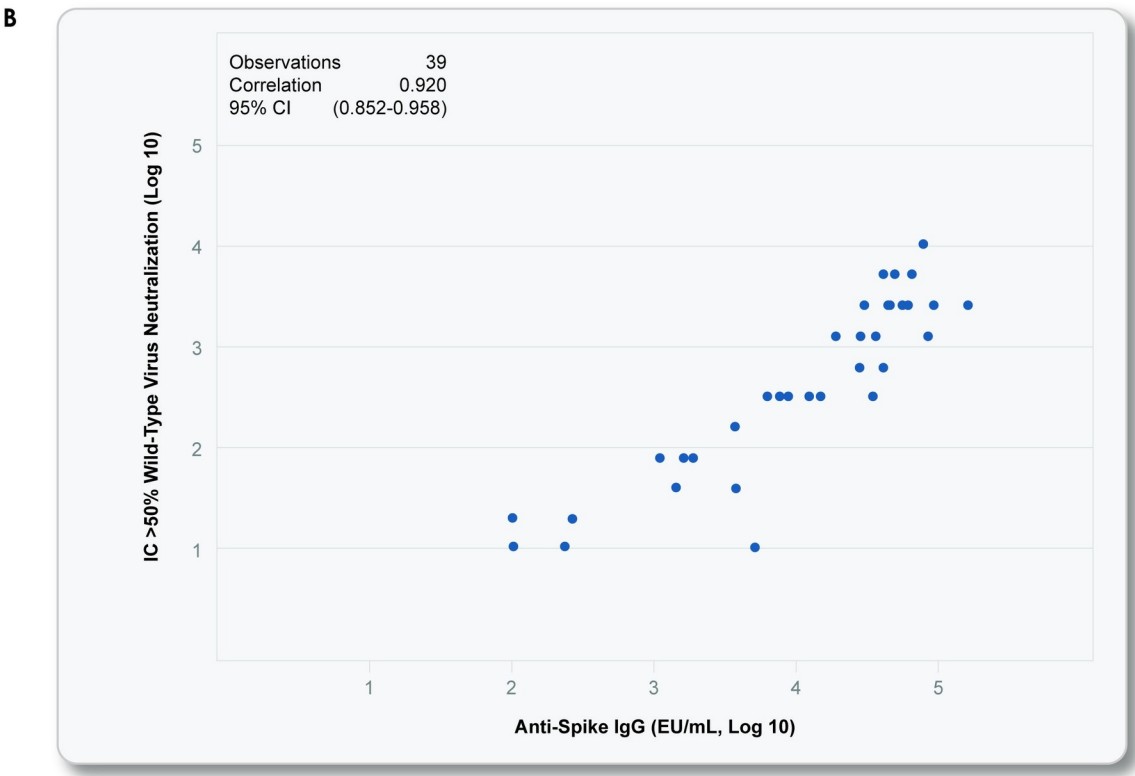

**Fig 4. Correlation between SARS-CoV-2 anti-spike IgG and neutralizing antibody responses.** Shown is a scattergram and correlation analysis of the neutralizing antibody titers and anti-spike IgG ELISA units for the 5-μg NVX-CoV2373 2-dose vaccine group (Group B)

for younger adults 18 to 59 years (Panel A) and older adults 60 to 84 years (Panel B). Pearson correlations were calculated with 2-sided 95% CIs. The correlation plots were inclusive of paired specimens by the 2 assays at the Day 21 time point following first vaccination, as well as for paired results at the Day 35 time point following second vaccination. (**A**) 5 μg + Matrix-M1 (Group B)– 18–59 years [footnote for Fig 4A] * NCT04368988 –phase 1. (**B**) 5 μg Matrix-M1 (Group B)– 60–84 years. CI, confidence interval; ELISA, enzyme-linked immunosorbent assay; EU, endotoxin unit; IC, inhibitory concentration; IgG, immunoglobulin G; SARS-CoV-2, severe acute respiratory syndrome coronavirus 2.

convalescent sera from a clinical case series of asymptomatic, outpatient, and hospitalized patients.

The safety and tolerability data from this study are also consistent with that gathered from over 14,000 participants [14–21], including over 4,300 participants who received the Matrix-M1 adjuvant [18–21], enrolled in previous noncoronavirus nanoparticle trials. These studies included children, pregnant women, and older adults with an age range from 5 months to 85 years of age and indicate that vaccines based on the nanoparticle/Matrix-M1 adjuvant technology have an acceptable safety profile to date.

Study limitations include the relatively short duration of safety follow-up to date and current lack of immune persistence data beyond the primary vaccination regimen time point assessments, but these data will accumulate over time.

Other limitations include restricted eligibility criteria that excluded advanced age participants (85 years and older) and persons with severe comorbidities and immunocompromised persons, although this phase 2 study was more permissive of inclusion of persons with clinically stable comorbidities than would normally be the case for a phase 2 study, given the pandemic situation to broaden the generalizability of the results. Another limitation was the lack of formal statistical inference to assess superiority or noninferiority of the 2 vaccine regimens.

There is not currently an established immune correlate of protection to establish the clinical significance of the immunogenicity results; however, clinical endpoint studies underway by Novavax and other companies for a variety of SARS-CoV-2 vaccine candidates have been demonstrating that multiple vaccines targeting immune responses directed against the spike protein of SARS-CoV-2 are demonstrating substantial initial vaccine efficacy with acceptable safety/tolerability profiles in adults and/or older adults against the vaccine strains included [6,7,13,22–25]. In settings where mass vaccination programs have been implemented for some of the first licensed vaccines, vaccine effectiveness studies have begun to demonstrate effectiveness estimates that are similar to clinical efficacy results from phase 3 studies [26,27]. The emergence of SARS-CoV-2 variants circulating in countries concurrently during some clinical efficacy studies have allowed some vaccines to report efficacy against emerging variants [13,22,28,29].

The data from this phase 2 study indicate that the Matrix-M1 adjuvanted rSARS-CoV-2 nanoparticle vaccine was highly immunogenic and well tolerated in younger and older participants.

## Supporting information

**S1 CONSORT Checklist.**
(DOC)

**S1 Text. Table A.** Toxicity grading scales for solicited local and systemic adverse events— modified from FDA toxicity grading scale for clinical abnormalities. **Table B.** Adverse events of special interest relevant to COVID-19. **Table C.** Potential immune-mediated medical conditions. **Fig A.** Vaccine regimens and key trial assessments. **Table D.** Overall summary of solicited and unsolicited adverse events following vaccination of SARS-CoV-2 rS with Matrix-M1

adjuvant in adult participants 18 to 84 years of age (Safety Analysis Set). **Table E.** Percentage of all participants (18 to 84 years) experiencing solicited local and systemic adverse events by symptom, vaccination dose, vaccine group, and maximum toxicity grade (Safety Analysis Set). **Table F.** Percentage of younger adults (18 to 59 years) experiencing solicited local and systemic adverse events by symptom, vaccination dose, vaccine group, and maximum toxicity grade (Safety Analysis Set). **Table G**. Percentage of older adults (60 to 84 years) experiencing solicited local and systemic adverse events by symptom, vaccination dose, vaccine group, and maximum toxicity grade (Safety Analysis Set). **Table H.** Unsolicited adverse events by system organ class and preferred term reported in 1% or more participants in any group through 35 days after first vaccination (Safety Analysis Set).
(DOCX)

## Acknowledgments

We thank all of the study participants who volunteered for this study. Editorial assistance on the preparation of this manuscript was provided by Phase Five Communications, supported by Novavax, Inc.

Disclosure forms provided by the authors are available with the full text of this article at the *PLOS Medicine* website.

We acknowledge the contributions of the 2019nCoV-101 Study Group: Mark Adams, Mark Arya, Eugene Athan, Ira Berger, Paul Bradley, Richard Glover II, Paul Griffin, Joshua Kim, Scott Kitchener, Terry Klein, Amber Leah, Charlotte Lemech, Jason Lickliter, Mary Beth Manning, Fiona Napier-Flood, Paul Nugent, Susan Thackwray, and Mark Turner. We also acknowledge the contributions to the Part 2 of the study by Cheryl Keech, the clinical lead when the Part 1 of the 2019nCoV-101 Study started.

## Author Contributions

**Conceptualization:** Neil Formica.

**Data curation:** Neil Formica.

**Formal analysis:** Neil Formica.

**Investigation:** Neil Formica.

**Methodology:** Neil Formica.

**Project administration:** Neil Formica.

**Writing – original draft:** Neil Formica, Raburn Mallory, Gary Albert, Michelle Robinson, Joyce S. Plested, Iksung Cho, Andreana Robertson.

**Writing – review & editing:** Neil Formica, Raburn Mallory, Gary Albert, Michelle Robinson, Joyce S. Plested, Iksung Cho, Andreana Robertson, Filip Dubovsky, Gregory M. Glenn.

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
