## [Editor Report · Decision Letter 0]

2 Apr 2021

Dear Dr Formica, 

Thank you for submitting your manuscript entitled "Evaluation of a SARS-CoV-2 vaccine NVX-CoV2373 in younger and older adults" for consideration by PLOS Medicine.

Your manuscript has now been evaluated by the PLOS Medicine editorial staff and I am writing to let you know that we would like to send your submission out for external peer review.

Please re-submit your manuscript within two working days, i.e. by April 6, 2021.

Kind regards,

Beryne Odeny

Associate Editor

PLOS Medicine

---

## [Decision Letter · Decision Letter 1]

10 May 2021

Dear Dr. Formica,

Thank you very much for submitting your manuscript "Evaluation of a SARS-CoV-2 vaccine NVX-CoV2373 in younger and older adults" (PMEDICINE-D-21-01479R1) for consideration at PLOS Medicine. 

Your paper was evaluated by a senior editor and discussed among all the editors here. It was also sent to independent reviewers, including a statistical reviewer. The reviews are appended at the bottom of this email and any accompanying reviewer attachments can be seen via the link below:

[LINK]

In light of these reviews, I am afraid that we will not be able to accept the manuscript for publication in the journal in its current form, but we would like to consider a revised version that addresses the reviewers' and editors' comments. Obviously we cannot make any decision about publication until we have seen the revised manuscript and your response, and we plan to seek re-review by one or more of the reviewers. 

We expect to receive your revised manuscript by May 31 2021 11:59PM. Please email us (plosmedicine@plos.org) if you have any questions or concerns.

We look forward to receiving your revised manuscript. 

Sincerely,

Beryne Odeny, 

PLOS Medicine

plosmedicine.org

1) Please revise your title to indicate that this is an implementation science study. Your title must be nondeclarative and not a question. It should begin with main concept if possible. For example, please place the study design ("A phase II randomized placebo-controlled trial,") in the subtitle (i.e., after a colon). 

2) Abstract summary - At this stage, we ask that you reformat your non-technical Author Summary. The Author Summary should immediately follow the Abstract in your revised manuscript. This text is subject to editorial change and should be distinct from the scientific abstract. The summary should be accessible to a wide audience that includes both scientists and non-scientists. Please see our author guidelines for more information: https://journals.plos.org/plosmedicine/s/revising-your-manuscript#loc-author-summary.

3) Abstract:

a) Please structure your abstract using the PLOS Medicine headings (Background, Methods and Findings, Conclusions).

b) Please combine the Methods and Findings sections into one section, “Methods and findings”. Please ensure that all numbers presented in the abstract are present and identical to numbers presented in the main manuscript text.

c) Please report your abstract according to CONSORT for abstracts: http://www.consort-statement.org/extensions?ContentWidgetId=562

d) Please quantify the main results (with p values in addition to 95% CI).

e) Please include a summary of adverse events if these were assessed in the study.

f) In the last sentence of the Abstract Methods and Findings section, please describe the main limitation(s) of the study's methodology.

4) Please complete the CONSORT checklist (CONSORT extension for early phase dose-finding clinical trials: https://www.equator-network.org/library/reporting-guidelines-under-development/reporting-guidelines-under-development-for-clinical-trials/ ). When completing the checklist, please use section and paragraph numbers, rather than page numbers.

5) Please provide the following as Supplementary Information files: trial design, conduct, oversight, and analyses are provided in the protocol and statistical analysis plan

6) How was race/ethnicity defined and by whom? Why was race/ethnicity considered important in this study and what it is believed to represent [eg, are differences being attributed to race/ethnicity?]

7) In the Methods and Results section:

a) Please provide 95% CIs and p values for estimates in the main text and tables

b) When a p value is given, please specify the statistical test used to determine it.

8) Figures and tables:

a) Please indicate in the figure caption the meaning of the whiskers in Figures

b) Please provide definitions for the following abbreviations: IC, AEs, EU/mL

9) Please replace "subject" with participant, patient, individual, or person.

10) Under the “Discussion” section, 1st paragraph (line 337, 342), the term "trend" is used to refer to a nonsignificant P value. The term trend should be used only when the test for trend has been conducted. Please revise accordingly.

Comments from the reviewers:

Reviewer #2: This is a Phase 2 trial reporting the immunogenicity and reactogenicity profile of a matrix-M1 adjuvanted spike protein SARS-Cov-2 vaccine. This is not clear from the title of the paper which just mentions "evaluation" of a "SARS-CoV-2 vaccine NVX-CoV2373". Most readers will not be familiar with this name for the vaccine. It would help if the title indicated this was a Phase 2 trial and also the vaccine format, eg as per the Phase I NEJM preliminary report. Can the authors clarify whether any of the 18-59 year olds in the current Phase 2 were also reported in the Phase 1 as the NEJM paper and this one both refer to the Phase1-2 trial with the Phase 2 component enrolling older adults. 

Abstract: The abstract gives no idea of the numbers of subjects or CIs around the key immunogenicity results. For example on page 2 line 41 it says that seroconversion rates in the wild type neutralisation assay are 100% in both age groups. However this is based on just 49 subjects across both age groups with no age breakdown given (Table S9) so the CIs around this estimate by age group will be quite wide. Also it is unclear how the subset tested for neutralising antibodies was selected. 

Introduction: Page 4 line 74 - what does "strong/high" levels mean in relation to a Th1 response? I could not find mention of "strong/high" in the NEJM paper to which this statement refers.

Fig 1 Consort diagram. It wasn't immediately clear to me what the 5/50 and 0/50 notation referred to. Can this be clarified in the text when there is the initial mention of the 5 study groups. Also from the supplementary appendix it appears that some of the groups will receive a third dose at 189 days which might merit a mention in the main text, eg discussion?. 

Page 7 line 119. Does "within 7 days" mean there was a daily electronic diary that had to be completed each day for 7 days or was the reactogenicity assessment just at day 7? 

Page 7 line 132 - what do the authors mean by the term "validated" assay and also on line 135 by a "qualified" assay? 

Page 8 line 140. It would be helpful for readers to know that the convalescent panel comprised 29 subjects with predominantly mild disease without having to look this up in the NEJM paper. 

Statistical analysis: Page 8. There are no power calculations shown to justify the sample sizes just the assertion that with at least 150 participants per groups "there was more than adequate power to detect differences between vaccine groups" . Can the authors provide a proper power calculation which must have been done to support the chosen trial size which was designed to compare schedules and doses to select the one to enter the Phase 3 trial. 

 Results: The long narratives reporting by age group and vaccine group the various immunogenicity and reactogenicity results make very difficult reading and are really hard to take in. Moreover there are no CIs nor numbers per group provided in the text nor are there any statements when comparing %s between groups whether any differences cited are likely to significant or not. Why relegate all the tables to the supplementary appendix as they are a much more efficient way of communicating the results and their robustness to readers and the text need only then just summarise the key messages from that tables. 

Unsolicited adverse events: page 15. I found this confusing to read as I think that some SAEs appear multiple times under different headings - at least I hope so! This section needs reorganising so that readers can clearly see the numbers of occurrences of the events mentioned, irrespective of what category they come under in the the protocol definition. 

Page 17: line 296 - this speculation about immunosenescence belongs in the discussion not the results.

Discussion Line 370. The authors state that the neutralising antibodies were similar to or exceeded those "in a small sample of convalescent sera from hospitalised patients". In fact looking at the NEJM paper in which information on this sample is provided it seems that there are only 4 sera among the 29 that are from hospitalised patients and they had a GMT of 7457 which vastly exceeds that shown in Table S9 for the 49 individuals aged 18-84 years with post dose 2 neutralising titres in both the low and high dose groups. Can the authors provide the evidence for their statement?. 

The authors say they have no conflicts of interest but I note that all are employed by Novavax. 

Reviewer #3: I appreciated the opportunity to review this paper on safety and immunogenicity of the NVX-CoV2373 vaccine in younger and older adults. Overall it is a clearly written paper which certainly makes a useful contribution to the evolving literature on COVID 19 vaccines. The methods are rigorous and well-described. Validated or qualified laboratory assays were used. The figures presenting the safety and immunogenicity data are clear.

I have the following comments:

1. Methods

a)please specify what placebo was used 

b)what was the follow up period for SAE reports presented here?

2. Results

a)the mean age of the 60-84 year old participants was approx 67 between groups. The authors should discuss that this study cohort does not reflect the elderly population most at risk of severe COVID. Please also provide data on co-morbidties in this group e.g. cardiovascular disease, COPD to show how these results can be extrapolated to the frail elderly. 

b)line 206 - do you mean' For vaccine participants who received placebo as the second injection, frequencies of solicited local adverse events were similar to participants who received a single dose of placebo'? 

c)please clarify whether grade 4 potentially life threatening events are counted as SAEs (line 189). It appears that ER attendance does not count as a SAE but hospitalisation does. These definitions could be included in the supplementary material. 

d) please comment on 1.3% seroconversion in placebo group (line 280). Was this due to lack of specificity in the assay or asymptomatic disease in the cohort? Was this confirmed by N protein seroconversion? 

e) Please specify if any paticipants developed COVID -19 during the period described

f)please specify when recruitment occurred (month, year)

g) It would be useful to see comparative statistics on ELISA/neutralising tires between first and second doses at each dosing regime and between ages.

3. Discussion

a) The authors should set the context for their findings by considering other deployed vaccines that have looked at immunogenicity and reactogenicity in adults of different ages. E,g, Gamelaya, CanSino, AZ, Moderna. E.g. in contrast to the authors' findings, the AZ vaccine is less reactogenic in elderly adults. 

b)The authors should acknowledge that in the absence of a correlate of protection, the significance of absolute binding or neutralising antibodies is unclear. T cell immunity may also be important but is not presented here. They should also refer to the press-released Novavax efficacy data to link how the observed seroconversion may translate into protection from disease. 

4. Figures

a) Fig 1 please show number of participants who received dose 1 and who went on to receive dose 2. Please also clarify 'early termination' - is this study withdrawal? Is this the same as 'discontinued intervention'?

5. Supplementary 

a)Please present percentages of adverse events together with absolute numbers in the tables

[LINK]

---

## [Decision Letter · Decision Letter 2]

26 Jul 2021

Dear Dr. Formica,

Thank you very much for re-submitting your manuscript "A phase 2 randomized placebo-controlled trial of different dose regimens of a SARS-CoV-2 recombinant spike protein vaccine (NVX-CoV2373) in younger and older adults" (PMEDICINE-D-21-01479R2) for review by PLOS Medicine.

I have discussed the paper with my colleagues and the academic editor and it was also seen again by xxx reviewers. I am pleased to say that provided the remaining editorial and production issues are dealt with we are planning to accept the paper for publication in the journal.

[LINK]

We look forward to receiving the revised manuscript by Aug 02 2021 11:59PM.   

Sincerely,

Beryne Odeny, 

Associate Editor 

PLOS Medicine

plosmedicine.org

Requests from Editors:

1) Please revise your title, include setting, and place the study design ("A phase II randomized placebo-controlled trial,") in the subtitle after a colon. For example, “Different dose regimens of a SARS-CoV-2 recombinant spike protein vaccine (NVX-CoV2373) in younger and older adults in USA: A phase 2 randomized placebo-controlled trial”

2) Author summary: please use bullet point(s) under the subheading, “What do these finding mean?”

3) In the abstract, please include the setting/ location of the trial.

4) Please include p-values in addition to 95% CIs in both the main text (including abstract) and tables.

5) Thank you for providing your CONSORTchecklist. Please replace the page numbers with paragraph numbers per section (e.g. "Methods, paragraph 1"), since the page numbers of the final published paper may be different from the page numbers in the current manuscript.

6) There are still instances of the term “subject” in the main text. Please replace "subject" with participant, patient, individual, or person.

7) The terms gender and sex are not interchangeable (as discussed in http://www.who.int/gender/whatisgender/en/ ); please use the appropriate term.

Comments from Reviewers:

Reviewer #3: Thank you for addressing my queries, I have no further comments.

[LINK]

---

## [Editor Report · Decision Letter 3]

13 Aug 2021

Dear Dr Formica, 

On behalf of my colleagues and the Academic Editor, Dr. Maheshi N. Ramasamy, I am pleased to inform you that we have agreed to publish your manuscript "Different dose regimens of a SARS-CoV-2 recombinant spike protein vaccine (NVX-CoV2373) in younger and older adults: A phase 2 randomized placebo-controlled trial" (PMEDICINE-D-21-01479R3) in PLOS Medicine.

PRESS

Sincerely, 

Beryne Odeny 

Associate Editor 

PLOS Medicine